# Edible Seaweeds Extracts: Characterization and Functional Properties for Health Conditions

**DOI:** 10.3390/antiox12030684

**Published:** 2023-03-10

**Authors:** Mariana Coelho, Ana Patrícia Duarte, Sofia Pinto, Hugo M. Botelho, Catarina Pinto Reis, Maria Luísa Serralheiro, Rita Pacheco

**Affiliations:** 1Departamento de Engenharia Química, Instituto Superior de Engenharia de Lisboa (ISEL), Av. Conselheiro Emídio Navarro 1, 1959-007 Lisboa, Portugal; 2BioISI—Biosystems & Integrative Sciences Institute, Faculty of Sciences, Universidade de Lisboa, 1749-016 Lisboa, Portugal; 3Research Institute for Medicines (iMed.ULisboa), Faculty of Pharmacy, Universidade de Lisboa, Av. Professor Gama Pinto, 1649-003 Lisboa, Portugal; 4Instituto de Biofísica e Engenharia Biomédica (IBEB), Faculdade de Ciências, Universidade de Lisboa, Campo Grande, 1749-016 Lisboa, Portugal; 5Departamento de Química e Bioquímica, Faculty of Sciences, Universidade de Lisboa, 1749-016 Lisboa, Portugal; 6Centro de Química Estrutural—Institute of Molecular Sciences, Universidade de Lisboa, 1749-016 Lisboa, Portugal

**Keywords:** *Eisenia bicylcis*, *Porphyra tenera*, *Fucus vesiculosus*, antioxidant activity, acetylcholinesterase, extracts, phloroglucinol derivatives

## Abstract

Seaweeds are popular foods due to claimed beneficial health effects, but for many there is a lack of scientific evidence. In this study, extracts of the edible seaweeds Aramé, Nori, and Fucus are compared. Our approach intends to clarify similarities and differences in the health properties of these seaweeds, thus contributing to target potential applications for each. Additionally, although Aramé and Fucus seaweeds are highly explored, information on Nori composition and bioactivities is scarce. The aqueous extracts of the seaweeds were obtained by decoction, then fractionated and characterized according to their composition and biological activity. It was recognized that fractioning the extracts led to bioactivity reduction, suggesting a loss of bioactive compounds synergies. The Aramé extract showed the highest antioxidant activity and Nori exhibited the highest potential for acetylcholinesterase inhibition. The identification of the bioactive compounds in the extracts allowed to see that these contained a mixture of phloroglucinol polymers, and it was suggested that Nori’s effect on acetylcholinesterase inhibition may be associated with a smaller sized phlorotannins capable of entering the enzyme active site. Overall, these results suggest a promising potential for the use of these seaweed extracts, mainly Aramé and Nori, in health improvement and management of diseases, namely those associated to oxidative stress and neurodegeneration.

## 1. Introduction

Seaweeds have been an important dietary constituent in Asian countries such as China, Japan, and Korea [1]. In some regions of the world, the population is growing to reach a level where food production may not be sufficient to feed the population [2], therefore the demand for alternatives, whether for human consumption or industrial processing, has increased in recent decades [3]. Seaweeds are considered an abundant, rich and sustainable marine source of macro and micronutrients, and an alternative to animal or even synthetic products [3]. Furthermore, several seaweed compounds have been reported to be bioactive, thus providing beneficial health effects [2], with their bioactive compounds showing the potential to be used as ingredients, both in functional foods and in health and food supplements.

Regarding macronutrients, seaweeds are a good source of sulfated polysaccharides (15–76% dry mass), proteins (5–47% dry mass), including all the essential amino acids, and lipids (1–5% dry mass), such as polyunsaturated fatty acids [3,4]. Concerning micronutrients, seaweeds are a good source of lipid and water-soluble vitamins, namely A, B1, B12, C, D and E and minerals (7–36% dry mass), such as calcium, iron, magnesium, copper, and iodine [3,4,5]. Seaweeds are also a source of secondary metabolites such as polyphenols [1,3,4]. Although some of these compounds, especially the latter, have reported beneficial health effects, like antioxidant, antimicrobial, anti-inflammatory, anti-cancer, anti-diabetic, anti-hypertensive, anti-hyperlipidemic, and anti-obesity effects [6], there is an immense variety of seaweeds. Numerous reports concern individual species, and there are several reviews about these organisms. The widespread belief that seaweed products have the potential to treat or prevent most human conditions has been rising. However, often there is lack of the proof of concept for the claimed effects and no association of a particular species to the bioactivities, which are then assigned to the whole group.

Seaweeds are taxonomically classified into three main groups, brown seaweeds (phylum *Ochrophyta*), red seaweeds (phylum *Rhodophyta*), and green seaweeds (phylum *Chlorophyta*) [7].

Seaweed colors are associated with its pigment, such as chlorophyll for green, phycobilin for red, and fucoxanthin for brown seaweeds [3]. The compound content of each seaweed depends on the species, but also on other geographical and environmental factors, such as substrate firmness, exposure to ice and waves, salinity, wave force, light and competition between seaweeds [8,9]. For this work, three seaweeds commonly used in foods or supplements were chosen to be studied, and their composition and biological activities were compared to report on their given potential for health improvement and management of diseases, especially those associated with oxidative stress.

*Eisenia bicyclis*, traditionally known as Aramé, is a perennial brown seaweed [10] distributed along the mid-pacific coastlines of Korea and Japan [10]. This seaweed is used industrially to extract sodium alginate, and for consumption by the population of East Asia in soups and salads [11,12]. The main bioactive compounds reported for Aramé are phenolic compounds, and polysaccharides [13,14]. The most abundant phenolic compounds are phlorotannins (eckol and other phloroglucinol derivatives), which were reported to have various biological activities, namely anti-diabetic and antioxidant activities [15]. The most abundant polysaccharides seen to be present in Aramé were fucoidan and laminarin [14]. On *Ecklonia cava* [16] and *Sargassum vulgare* [17], other brown seaweeds, these polysaccharides were reported to have antioxidant activity and anti-inflammatory activity [16,17]. Nevertheless, to the best of our knowledge, there is no research about these type of activities for Aramé fucoidan.

*Fucus vesiculosus*, known as bladderwrack [18], is an edible brown seaweed of the rocky bottoms of the northern temperate coastal areas [19], used to make beverage infusions, in cooked dishes and soups, or sprinkled in salads [20]. In traditional medicine Fucus is used due to its high iodine content to treat gout and aid weight loss [21]. This seaweed has bioactive compounds such as phenolic compounds [22], and sulfated polysaccharides [23]. It also contains proteins, minerals, vitamins, fatty acids, sterols, dietary fiber, and iodine [20,24]. The phenolic compounds mostly present in this seaweed are phlorotannins [25], which have reported activities such as antioxidant [18], anti-diabetic, anti-inflammatory, anti-cancer, anti-obesity, anti-lipidemic, and anti-hypertensive [20,26,27,28]. Also, Fucus polysaccharides are fucoidan and laminarin [20]. The latter was reported to have anti-inflammatory, anticoagulant, antioxidant, anti-cancer, and hypolipidemic activity [29,30,31], but again for fucoidan from Fucus there are no reports.

*Porphyra tenera*, known as Nori, is a red seaweed considered the most valuable maricultured seaweed in the world, as it is one of the most popular edible seaweeds used in sushi and soups [32,33]. It is known to have major bioactive compounds phenolic compounds, sulfated polysaccharides, and peptides [34,35]. It is also a source of dietary fiber, essential fatty acids, vitamins, and minerals [36]. In Nori, phlorotannins were identified and described to exhibit antioxidant activity and protection against UV light [37,38]. Other bioactive compounds present in this seaweed are sulfated polysaccharides, called porphyrans, which are reported to have hypolipidemic, anti-cancer, and anti-inflammatory activities [39]. Other polysaccharides reported in red seaweeds, such as cellulose, xylans, and manans, are not water-soluble [40].

In this work, a novel approach was used. Three of the most consumed edible seaweeds, Aramé, Fucus, and Nori, were matched for the composition and biological activities associated to the aqueous extracts of the seaweeds. Though there are various reports about Aramé [13,14], Fucus [22,23,24], and Nori [34,35], individually there are no studies comparing their composition and beneficial health effects.

With this in mind, we herein report the characterization of aqueous extracts from the three seaweeds, obtained under the same conditions, and fractions of these extracts, enriched in different bioactive compounds classes from the three seaweeds. Their biological activities, namely the antioxidant activity and enzyme acetylcholinesterase (AChE) inhibition were investigated and compared in order to evaluate their potential against oxidative stress. In this work, reported for the first time is the capacity of Nori extracts to inhibit acetylcholinesterase, an enzyme associated to gastrointestinal motility and neurodegenerative diseases, as Alzheimer’s disease (AD). The comparison of the seaweeds extracts guided the association between the exhibited activities and their composition in bioactive compounds, supporting future exploration of targets for application. Additionally, the results herein reported may encourage the development of novel and natural products with the incorporation of these seaweeds into the diet, supplements, or functional foods, particularly to prevent oxidative stress. Oxidative stress is often associated to several diseases, such as cardiovascular diseases, metabolic conditions, and neurodegenerative disorders.

## 2. Materials and Methods

### 2.1. Seaweeds

*Porphyra tenera* was purchased from Flavers-International Flavours Shop^®^ (Blue Dragon line, B#JS2039J01). While *Fucus vesiculosus* was collected from Tagus River (38.7822 N, 9.0913 W). The dry *Eisenia bicyclis* seaweed was purchased in a commercial surface from the Seara brand (B# T20220405, expiration date April 2022), as previously described in [13].

### 2.2. Chemical

All reagents and solvents were of analytical grade unless otherwise specified and used without further purification. Roswell Park Memorial Institute (RPMI-1640), Dulbecco’s Modified Eagle Medium (DMEM), trypsin and glutamine from Biowhittaker^®^ Lonza. Fetal Bovine Serum (FBS) from Biowest, phosphate-buffered saline (PBS) were obtained from Corning (Corning, NY, USA). Antibiotic Antimycotic Solution 100 × (10,000 U/mL penicillin, 10 mg/mL streptomycin, and 25 μg amphotericin B/mL), reagent Folin & Ciocalteu, sodium acetate, 2,2-diphenyl-1-picyl-hydroxyl (DPPH), acetylcholinesterase (AChE) (149 U/mg solid, 241 U/mg protein), and acetycholine iodide (AChI) were obtained from Sigma^®^Aldrich (St. Louis, MO, USA). Calcium Carbonate, Concentrated Sulfuric Acid from Merck. Phloroglucinol from Aldrich^®^ chemistry. Polygalacturonic acid and 5-5′-Dithiobis (2-nitrobenzoic acid) (DTNB) were purchased to Alfa Aesar (Ward Hill, MA, USA). Phenol and (4,5-dimetylthiazol-1-yl)-2,5-diphenyltetrazolium (MTT) were obtained from VWR (Radnor, PA, USA). Citric acid and Magnesium Chloride-6-hydrate were obtained from Riedel-de Haën (Seelze, Germany). Sodium Chloride was obtained from Panreac (Glenview, IL, USA).

### 2.3. Preparation of Seaweed Extracts

The dried and milled biomass of the seaweeds was used to prepare aqueous extracts. As the biomass of Fucus vesiculosus was collected in nature, it was subjected to an extensive washing procedure and further dried in an Heto^®^ PowerDry LL3000 freeze dryer.

For the preparation of a dry mass of Nori aqueous extract, the biomass of *Porphyra tenera* was mixed with distilled water (20 g/L) and the suspension was autoclaved at 121 °C for 15 min. The suspension was filtered, frozen at −20 °C and freeze dried, to obtain the dry extract (50% g/g yield). The same procedure was performed to obtain a dry mass of Fucus aqueous extract, using 10 g/L biomass of *Fucus vesiculosus* (38% g/g yield). The procedure for obtaining a dry mass of Aramé aqueous extract using 33 g/L biomass of *Eisenia bicylis* (66% g/g yield) was already reported in a previous publication from our group [13].

### 2.4. Extract Fractioning by Solid Phase Extraction (SPE)

Solutions of the Aramé, Nori, and Fucus extracts were prepared by resuspending the extract dried mass in water to a concentration of 30 mg/mL, 15 mg/mL and 11.8 mg/mL, respectively, loaded into Sep-Pak C18 Plus Short Cartridge (360 mg sorbent per cartridge, 55–105 µm particle size, 50/pk), which had been pre-conditioned with methanol followed by water. One mL of the extract solution was loaded to the cartridge and then water 83 mL) and methanol (5 mL) were added, the procedure was repeated at least 3 times per cartridge, a total volume of 10 mL was used per extract solution. The fractions collected in water were frozen at −20 °C, freeze dried and named Aramé H_2_O, Nori H_2_O, and Fucus H_2_O. The fractions collected in methanol were evaporated and named Aramé MeOH, Nori MeOH, and Fucus MeOH.

### 2.5. Extracts and Fractions Characterization

The extract and fraction dried mass, obtained in Section 2.3 and Section 2.4, was dissolved in water to prepare solutions that were used for further quantifications and the biological activities assays, except for the cytotoxicity assays where the dried mass was dissolved in cells growth medium.

#### 2.5.1. Quantification of the Total Phenolic Content (TPC)

The total phenolic content was determined according to the Folin-Ciocalteu method [41], and the results were expressed as mg of phloroglucinol equivalents (PGE) per mg of dry mass (phloroglucinol 0–0.06 mg·mL^−1^; R^2^ = 0.95), as the mean of triplicates. Briefly, 1350 μL water, 30 μL Folin-Ciocalteau reagent, 30 μL sample (extract, fraction, or standard phloroglucinol) solution, and 90 μL Na_2_CO_3_ (2% *w*/*v*) were kept in an orbital shaker for 1 h at 4 °C and, afterwards, the absorbance was measured at 760 nm in an UV-Vis Shimadzu spectrophotometer against a blank containing water instead of sample.

#### 2.5.2. Quantification of the Total Proteins

For the quantification of total proteins, the 2-D Quant Kit from GE Healthcare^®^ was used and the procedure was followed according to manufacturer instruction [42]. Bovine serum albumin (BSA) was used as standard to obtain a calibration curve (BSA 0–40 µg; R^2^ = 0.98) and the results were expressed in mg total proteins/mg dry mass, as the mean of triplicates.

#### 2.5.3. Quantification of the Total Polysaccharides

The concentration of polysaccharides in the extracts and fractions was determined according to the phenol-sulfuric acid method, as described in [43]. The results were expressed as mg of polygalacturonic acid equivalents (PE) per mg of dry mass (polygalacturonic acid 0–0.2 mg·mL^−1^; R^2^ = 0.99), as the mean of triplicates. Briefly, 50 μL sample (extract, fraction, or standard polygalacturonic acid) solution, 150 μL concentrated sulfuric acid water, and 30 μL phenol solution (5% *w*/*v*) were incubated for 5 min at 90 °C. After cooling the absorbance was measured at 490 nm in a microplate reader TECAN Sunrise.

### 2.6. Biological Activities

#### 2.6.1. In Vitro Safety in Caco-2 and Hep-G2 Cells

Hepatocellular carcinoma cell line Hep-G2 (ECACC#85011430) and colorectal adenocarcinoma cell line Caco-2 (ECACC#86010202) were cultured in DMEM and RPMI-1640 medium, respectively, and supplemented with inactivated FBS 10% (DMEM) or 20% (RPMI-1640), antibiotic-antimycotic (100 U/mL penicillin-streptomycin and 0.25 μg amphotericin B), and 2 mM L-glutamine at 37 °C in an atmosphere with 5% CO_2_. The medium was changed every 48–72 h, and cells were harvested before reaching confluence using PBS and 1x trypsin, and grown in the supplemented medium in 96-well microplates in an incubator with 5% CO_2_ at 37 °C, until reaching 100% confluence.

These cells lines were used because the seaweed extracts are a food product or food supplement. This type of food goes to intestine and liver, so the aim was to evaluate the cytotoxicity of the methanol fractions.

For the in vitro evaluation of the cytotoxicity of the extracts and fractions, the 3-(4.5-dimethylthiazol-2-yl)-2.5-diphenyltetrazolium bromide (MTT) method, described in [44] was used. The cell viability was evaluated after 24 h incubation, with 100 µL solutions of the extracts and fractions at the concentration of 0.5 and 1 mg dry mass/mL in growth media. After incubation, the solutions were replaced by 100 µL of 0.5 mg/mL MTT solution in culture medium and incubated at 37 °C, 5% CO_2_ for 2 to 4 h. The formed formazan crystals were dissolved in 200 µL of methanol and the absorbance at 595 nm was registered against 630 nm (reference wavelength). For each solution, the percentage of growth inhibition/cytotoxicity was evaluated considering 100% of viability for the absorbance of the control (cells incubated in the same conditions solely in growth media).

#### 2.6.2. Antioxidant Activity

The antioxidant activity of the solutions of extract and fraction dry mass was measured using an adaptation of the DPPH method described in [44]. To 1 mL of 0.002% *w*/*v* DPPH solution in methanol, 25 μL of sample solution was added and incubated for 30 min at room temperature. Afterwards, the absorbance of the mixture was measured at 517 nm and the percentage of antioxidant activity (%) was determined using Equation (1), where Abs 517 nm control is the absorbance at 517 nm of the blank DPPH solution with water instead of sample solution and Abs 517 nm Sample is the absorbance at 517 nm of the sample solution. The assays were carried out in triplicate.
(1)$$\left(\%\right)=100\times \frac{Abs_{517\;nm\;control}-Abs_{517\;nm\;sample}}{Abs_{517\;nm\;control}}$$

For the Aramé extract, which showed the highest antioxidant activity, the EC_50_ value was also calculated. EC_50_ is the concentration of the extract showing 50% of DPPH-free radical scavenging activity, calculated by plotting the antioxidant activity for different concentration of the Aramé extract solutions.

#### 2.6.3. AChE Inhibitory Activity

The inhibition of acetylcholinesterase (AChE) enzymatic activity was measured using the Ellman’s colorimetric method with some alterations [44]. Briefly, 325 μL of 50 mM Tris–HCl buffer (pH 8), 100 μL of the extract solution, and 25 μL of AChE (0.1 U/mL) in 50 mM Tris–HCl buffer pH 8 were incubated for 15 min. Subsequently, 75 μL of acetylthiocholine iodide (AChI) (0.023 mg/mL) and 475 μL of 3 mM 5,5′-dithiobis(2-nitrobenzoic acid) (DTNB) in Tris–HCl buffer (pH 8) containing 0.05 M NaCl and 0.021 M MgCl_2_ were added to initiate the reaction. The initial rate of the enzymatic reaction was quantified by measuring the absorbance at 405 nm for 5 min (V[compound]). A control reaction was carried out using water instead of the extract solution, and this initial rate was considered 100% of the enzymatic activity, Vcontrol. The percentage of AChE inhibition (I) for the extracts was determined as the ratio of V[compound] and Vcontrol. All the assays were carried out in triplicate. The concentration of the extracts used was 1 mg/mL.

### 2.7. LC-HRMS/MS Extract Analysis

For the LC/HRMS analysis, a LiChrospher^®^ 100 RP-18 (5 μm) LiChroCART^®^ 250-4 mm column and a mobile phase constituted by a binary system of formic acid 0.1% MeOH at a rate of 1 mL/min. The method used was as follows: 0 min 80% formic acid 20% MeOH; 20 min 20% formic acid 80% MeOH; 25 min 20% formic acid 80% MeOH; 30 min 80% formic acid 20% MeOH. High resolution mass spectra were acquired using negative ESI mode because the goal was to identify phenolic compounds that are majority found in this mode. The results were analyzed by the mass higher than 100 (*m*/*z*) and intensities around 100%. All the extracts were injected with a concentration of 1 mg/mL. Mass spectra were acquired, in the range of 120–1000 *m*/*z* and the mass spectrometer parameters were adjusted to optimize the signal-to-noise (S/N) ratio for the ions of interest. Briefly, the flow rates of nebulization and auxiliary gas (nitrogen) were 40 and 20 arbitrary units, the capillary temperature was set at 250 °C and the collision energies at 40 and 45 eV. For the analysis of the mass spectrometry results, DataAnalysis software developed by Bruker^®^ (Darmstadt, Germany) was used.

### 2.8. Data Analysis

The software used for treatment was Microsoft^®^ Excel (Microsoft Office 365) and the results were expressed as average ± standard deviation. Additional analysis of variance was carried out using one-way ANOVA for values comparison, difference between mean values were considered significant when *p* < 0.05. 

## 3. Results

### 3.1. Extracts and Fractions Characterization

Extracts were prepared by first performing a water extraction from Arame, Nori, and Fucus biomass. The extracts were fractionated using SPE in water and methanol to obtain MeOH and H_2_O fractions for each extract. Methanol, according to ICH Q3, is limited to 3000 ppm per day (Class 2) due to its inherent toxicity, and therefore it was completely evaporated to obtain a dried mass of the fractions [45]. For comparison, the same procedure was performed both for the extract and water fraction.

#### 3.1.1. Quantification of the TPC

The results of the phenolic quantification of the Arame, Nori, and Fucus extracts, as well as its MeOH and H_2_O fractions, are shown in Figure 1. The extract showing higher TPC was Aramé, with 0.062 ± 0.005 mg PGE/mg dry mass [13]. However, the methanol fraction of Aramé showed the highest TPC per mg dry mass amongst all the analyzed samples. In the case of the water fraction of Nori, it was seen to have a low TPC below the detection limit. This behavior suggests a different composition of phenolic compounds for each seaweed extract, appearing to be mostly hydrophobic in the Nori extracts, and with the Aramé and Fucus extracts also having water-soluble phenolic compounds.

#### 3.1.2. Quantification of Total Proteins

Proteins were found at low concentration in the Fucus extract (0.0074 ± 0.0004 mg/mg dry mass), being below the detection limit in all other samples.

Although seaweeds are considered a rich source of proteins, their aqueous extracts are not, as already reported for the Aramé extract [13]. This may be caused by the complex cell walls hampering protein extraction from the crude biomass [46] but also by the temperature used for biomass extraction, which causes protein denaturation.

#### 3.1.3. Quantification of Total Polysaccharides

The results for polysaccharides quantification are shown in Figure 2, reported as mg of polygalacturonic acid equivalents (PE)/dry mass. In the case of the extracts, Nori showed the highest quantity and Aramé the lowest. Regarding Nori and Aramé extract fractions, these showed highest content of polysaccharides per mg dry mass. In the case of Nori, the fractions of methanol were enriched with these compounds, as were both fractions of Aramé, containing higher content in polysaccharides than the extract per mg dry mass. This was not the case with Fucus samples.

As mentioned, sulfated polysaccharides such as fucoidan have already been found in brown seaweeds and others, such as laminarin [47]. In the case of red seaweeds, porphyran is present in the cell walls of the red macroalgae *Porphyra* [48].

### 3.2. Biological Activities

#### 3.2.1. In Vitro Safety of Seaweed Extracts in Caco-2 and Hep-G2 Cells

As the seaweed extracts and their fractions may contain some compounds at very high concentration, the safety of these was addressed to eliminate concerns regarding hepatic toxicity or intestinal damage associated either to the dose or to the type of compounds present. Liver (Hep-G2) [49] and intestinal epithelial cells (Caco-2) [50] were used to evaluate the safety of the extracts and fractions for consumption. Hep-G2 cell line is widely accepted by regulatory agencies for medicines and food supplements to assess liver toxicity. The same applies for the Caco-2 cell system, a well characterized intestinal in vitro model with morphologic resemblance to intestinal epithelia.

According to the literature and ISO 10993, extracts or mixtures of compounds are considered not to be toxic to humans if its IC_50_, the concentration of extract to reduce 50% cell viability, is above a concentration of 0.1 mg/mL [51].

As can be seen in Figure 3 and Figure 4, both liver (Hep-G2) and intestine (Caco-2) cell viability against extract and fraction concentrations of 0.5 and 1 mg/mL, during 24 h, was always above this threshold. Thus, it can be concluded that the seaweed extracts and fractions obtained are not cytotoxic, which was relevant to evaluate before proceeding to other stages of the work. For Fucus only the extract was tested due to the consistent low compounds content in the previous trials.

#### 3.2.2. Antioxidant Activity

The antioxidant activity of the seaweed extracts and fractions are shown in Table 1. For the Aramé extract, the antioxidant activity for 0.25 mg/mL was 65 ± 3% [13], and an EC_50_, the concentration of the extract for achieving 50% antioxidant activity, of 0.174 mg/mL was obtained. Nori samples showed the lowest antioxidant activities.

For Aramé and Fucus, the antioxidant activity of the seaweed extracts was significantly higher than for the corresponding fractions, although, as seen previously, some of these fractions had higher TPC or polysaccharides per mg of dried mass relative to the extract. Evidently, the synergy of compounds in the extract mixture is an important feature of the exhibited antioxidant activity, as already seen in other cases [52]. As a result, only the extracts were analyzed for their capacity to inhibit acetylcholinesterase (AChE) enzyme.

#### 3.2.3. AChE Inhibitory Activity

AChE enzyme inhibition by the extracts was evaluated and the results are shown in Figure 5 for 1 mg dry mass/mL solutions of the extract. Though all the extracts showed a mild capacity to inhibit this enzyme, the highest value of inhibition was obtained for the Nori extract (28 ± 2%). Therefore, this suggests that the compounds in the Nori extract may be seen as a promising natural option for increasing gastrointestinal motility, when included in diet, or to ameliorate neurodegenerative disorders, as AD, as AChE inhibition is the target for pharmacological treatment of this disease.

### 3.3. LC-HRMS/MS Extracts Characterization

The characterization of the compounds present in the extracts was performed using liquid chromatography-high resolution tandem mass spectrometry (LC-HRMS/MS).

The chromatographic profiles of the samples were compared, and between the extracts at the same concentration some of the *m*/*z* peaks differed in intensity, and other peaks were only present in some of the extracts. The chromatograms were obtained in positive and negative modes, but only the negative mode is presented here (Appendix A). The compounds were tentatively identified based on the MS fragmentation patterns, literature sources [8,53,54], molecular masses, and predicted molecular formula (Table 2). Assuming that in phlorotannins with ether bonds, such as phloroethols or eckols, fragmentation can occur on either side of the ether bond of the attached phloroglucinol units, the expected MS/MS spectra of these compounds often present [M-H]^−^ at *m*/*z* 125, 141, and 110, corresponding to phloroglucinol, tetrahydroxy benzene, or resorcinol, as well as their combination with additional phloroglucinol, for example *m*/*z* 233, corresponding to two phloroglucinol units [53,54]. In the case of eckols, due to the presence of dibenzodioxin structures, the MS/MS spectra of compounds may present an [M-H]^−^ at *m*/*z* 261 corresponding to fragmentation of the ether bonds linking these structures to phloroglucinol units. All these [M-H]^−^ 125, 139, 111, 233, and 261 were seen in the MS fragmentation pattern for compounds *m*/*z* 575 and 596, and therefore these two compounds were tentatively assigned as eckol derivatives.

The *m*/*z* 265 was tentatively identified as a fuhalol derivative, corresponding to a tetrahydroxy benzene and its combination with phloroglucinol [53,54].

Finally, considering the MS fragmentation patterns of *m*/*z* 325, 311, and 339, and that in fucols, having only C-C bonds between its phloroglucinol units, is more likely to occur cross-ring cleavages [53,54], as well as their combinations with additional phloroglucinol, (e.g., [M-H]^−^ at *m*/*z* 165), and/or with water moieties (e.g., [M-H]^−^ at *m*/*z* 183), it is suggested that these phloroglucinol derivatives could be fucol type phlorotannins.

A heatmap representing the variation of the intensity of the tentatively assigned compounds between the extracts was also obtained to better understand the differences between the extracts (Table 2). In the extracts, several phlorotannins derivatives, such as eckol, fuhalol, and other phloroglucinol derivatives, were tentatively identified. Additionally, in the Fucus extracts, citric acid and glycidil compound were identified by comparing with those previously identified in our previous work [8,55].

Regarding phlorotannins, amongst the extracts, the Aramé extracts showed the highest intensity peak of *m*/*z* 575, and this was tentatively identified as an eckol derivative. This and other eckol derivatives were not detected in the other extracts. The Nori extract showed the highest intensity peaks for lower molecular weight phlorotannins, tentatively assigned as fucol type phloroglucinol derivatives, *m*/*z* 325, 311, and 339. These had lower intensity in the Aramé extract and were not detected in the Fucus extract. In the case of the Fucus compounds, the highest intensity was detected with *m*/*z* 265, proposed as a fuhalol derivative. This compound was also seen to be present in other Fucus extracts [8,53].

As previously seen, the highest activities were dependent on the extract; antioxidant activity was better for the Aramé extract and the Nori extract showed the highest capacity to inhibit AChE. This seems to indicate that higher molecular weight compounds, suggested as eckol derivatives in the Aramé rich extract, may be important for antioxidant activity. These, synergistically with lower molecular weight fucol type phloroglucinol derivatives, may explain the highest antioxidant activity achieved in this extract. On the other hand, these lower weight phlorotannins may well be also associated to the AChE inhibitory activity of the Nori extract. These smaller structures will be more prone to fit inside the enzyme’s active site. It is well known that AChE is inhibited by compounds having phenolic moieties in their structure due to the establishment of π–π interactions at the active site [56] and this may be the case for phlorotannins. However, the unit of phlorotannins, phloroglucinol, was used as standard and assayed for AChE inhibition and it was seen that 0.1 mg/mL solution showed 15 ± 2% inhibition, which is considered very small.

## 4. Discussion

As mentioned, seaweeds have been an important part of folk medicine and a source for dietary intake in Asian countries, which have been rapidly expanding globally in previous years. The demand for seaweeds has increased in proportion to the renewed interest in sustainable and alternative ways to provide sufficient healthy food for the growing global population. Seaweeds thus appear to contribute to the possibility of reaching the goal set by the United Nations for sustainable development, as several cultivation plants are addressing the sustainable cultivation of seaweed biomass across Europe [7,57,58]. Seaweed growth is not very demanding. They do not require fertilizers and their biomass captures carbon dioxide, having a negative carbon footprint. Their growth rate is higher than plants and they are less likely to be infected with pests and other diseases [59].

The increased interest in seaweeds was also significantly influenced by their claimed health benefits, which are often associated with the presence of several bioactive compounds. However, due to differences amongst several species and the variety of compounds present in these organisms, most of their bioactive compounds are yet to be identified and associated to their potential effects, hindering the proof–of-concept necessary for their application, either in therapeutic areas or as functional foods.

This work aims to increase knowledge about the future trends and perspectives for the application of three of the most common edible seaweeds: Aramé, Nori, and Fucus. A novel approach to address this issue was used. The seaweed extracts and their compounds, separated in enriched fractions, were characterized and compared in terms of composition and exhibited biological activities.

The total phenolic content, proteins, and polysaccharides content of the extracts and corresponding fractions were evaluated, and it was seen that fractionating the compounds present in the extracts had a major impact on the extract’s composition and in the exhibited biological properties.

Regarding the DPPH antioxidant activity, both the Aramé and Fucus extracts exhibited good antioxidant activity. The Aramé extract demonstrated the strongest antioxidant activity, and the opposite was seen for the Nori extract. Additionally, a clear reduction was seen in the antioxidant activity of the fractions, although some of the fractions, relative to the extract, were richer in compounds, such as phenolic compounds, often associated to high antioxidant activity [60], and polysaccharides [31]. This type of behavior indicates that bioactivity depends on the combinations of compounds present in the extract, and the synergistic effect of the mixture of compounds is evidently missing in the fractions causing poorest performance for the latter.

Mass spectrometry tentative identification of compounds present in the extracts showed that the Aramé extract, exhibiting the highest antioxidant activity, contained a combination of eckols, fuhalol, and other phloroglucionol derivatives of lower molecular weight. As shown by Brand-Williams and colleagues [61], there is a correlation between the interaction with DPPH radical antioxidant capacity and the structural conformation of the compounds tested. The capacity for DPPH scavenging, associated to phlorotannins, may be correlated to the extent of electron donating groups, such as the hydroxyl group, especially at the ortho or para position, occurring in the phloroglucinol unit. Additionally, phlorotannins with a lower polymerization degree, such as eckol and phloroeckol, show an increased antioxidant activity relative to higher molecular weight, such dieckol and 8,8’-bieckol, as higher polymerization condenses electron donating groups [62]. As for the analysed extracts, but mainly for the Aramé extract, a combination of lower molecular weight phlorotannins was identified, and therefore in accordance with the previous observations. The Nori extract was seen to contain lower TPC and a less significant array of phlorotannins, and both of these circumstances affected the antioxidant capacity.

Regarding the AChE activity, overall, the extracts showed milder inhibitory capacity, with this being the first report for a Nori extract. The Nori extract showed the best results for AChE inhibition, therefore its major bioactive compounds, tentatively identified as smaller sized fucol type phloroglucinol derivatives, could be suggested to be further explored for its potential to improve gastrointestinal motility and in the treatment/prevention of Alzheimer’s disease (AD). AChE is an enzyme located in the neuromuscular junctions and in the neurosynaptic gaps [63]. AChE inhibition may improve digestion [64], and there is also clinical evidence that the inhibition of AChE activity is an effective therapeutic target for the management of AD [65,66]. The inhibition of AChE increases the levels of ACh in synaptic cleft, which attenuates the cholinergic deficit associated to AD and improves cognition and memory function [67]. Also, it is reported that dementia is associated with poor nutrition, so it is suggested that the use of small size phlorotannins from seaweed extracts, as Nori extract phloroglucinol derivatives, could be effective both to inhibit the enzyme, improving symptoms in AD management, and a safe diet complement [68]. Oxidative stress has also been implicated in AD, because brain cells are predisposed to free radical attack due to their content and inability to synthesize antioxidant enzymes [69,70], which can lead to free radical attack at cell biomolecules [71]. Oxidative stress is associated to other chronic and degenerative diseases, such as cancer [72], cardiovascular associated diseases [73], multiple sclerosis [74], and rheumatoid arthritis [75]; as a debilitated antioxidant system within the organism may affect the elimination of reactive species formed during cell metabolism.

Thus, considering the results presented in this work, as the bioactive compounds from seaweed extracts demonstrated strong antioxidant activity, especially the mixture of eckols, fuhalol, and other phloroglucinol derivatives from brown seaweed Aramé, it is projected that the inclusion of these extracts or bioactive compounds in either a healthy diet, supplements, or functional foods is prone to improve health conditions.

The extracts did not show cytotoxicity in intestinal Caco-2 and liver Hep-G2 cell lines, which was expected as these seaweeds have been used this way in the diet for many years. The bioactive compounds incorporation into upcoming supplements or functional foods requires supplementary validation steps according to international standards. Among these, safety tests should be performed in normal cell lines, as the cell lines, Caco-2 [76] and Hep-G2 [49] used in this work have the metabolic profile characteristic of immortalized cells.

## 5. Conclusions

Seaweeds are a promising source of bioactive compounds, but their composition and associated biological activities may vary depending on the species and other attributes. Considering that most of these issues continue to be underexplored, the present study aimed to elaborate a scientific comparison between extracts of the three of the most consumed seaweeds Aramé, Nori, and Fucus, regarding the commonly claimed qualities and biological activities associated to seaweeds.

It was seen that these seaweed extracts might have therapeutic potential against oxidative stress and neurological disorders, such as Alzheimer’s disease. It was additionally seen that, for all seaweeds, the mixture of bioactive compounds, obtained by extraction in hot water—as often used for consumption—had the most promising antioxidant activity, relative to extract fractions containing the separated compounds. The Aramé extract, seen to be a mixture of phlorotannins, tentatively assigned as eckols, fulahol, and other phloroglucinol derivatives, showed the highest antioxidant potential, therefore promising against oxidative stress associated conditions. Results also showed that the red seaweed Nori extract, containing smaller sized phlorotannins, showed the lowest antioxidant activity. On the other hand, the Nori extract demonstrated the highest AChE inhibitory capacity when compared with the other seaweeds; therefore, Nori bioactive compounds emerge as promising to improve digestion and complement AD management.

In conclusion, given the interesting outcomes with the extract’s bioactive compounds association to health improvement and management of diseases, mainly those associated with oxidative stress and neurodegeneration, future trends and perspectives are to persist towards the validation of the extract’s capacity to be used as dietary supplements or functional food, pursuing the development of seaweed-based food or novel and natural products with the incorporation of these seaweeds.

## Figures and Tables

**Figure 1 antioxidants-12-00684-f001:**
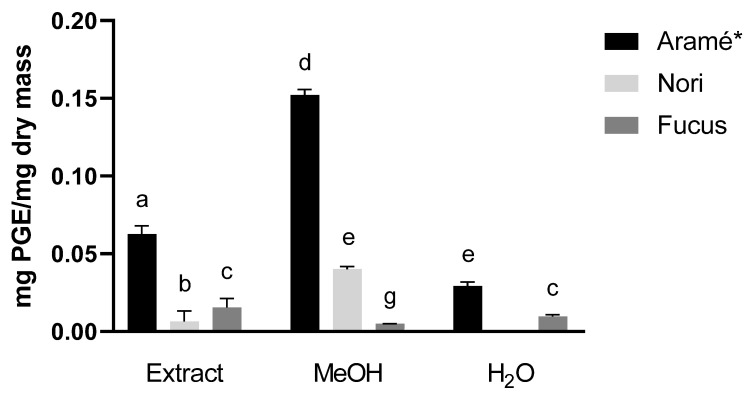
Total phenolic content per amount of dry mass for extracts and fractions purified water (H_2_O) and methanol (MeOH). Letters a–g correspond to values that are statistically different between the samples under study (*p* < 0.05). * results reported in [13].

**Figure 2 antioxidants-12-00684-f002:**
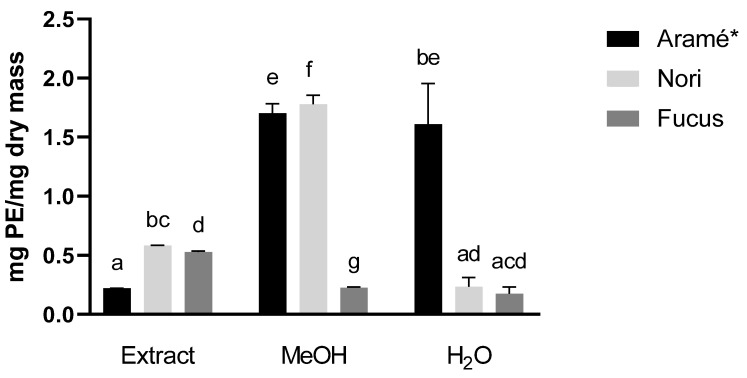
Polysaccharides content per amount of dry mass for extracts and fractions. Letters a–g correspond to values that are statistically different between the samples under study (*p* < 0.05). * results reported in [13].

**Figure 3 antioxidants-12-00684-f003:**
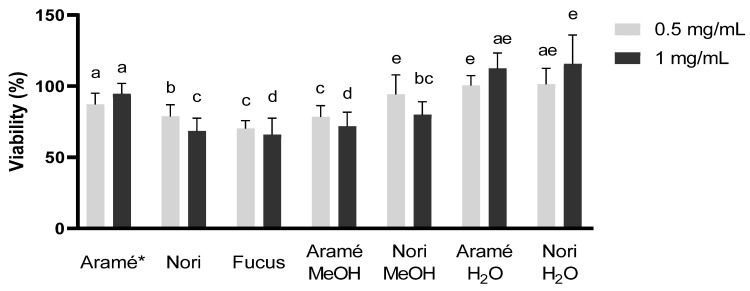
Caco-2 cell cytotoxicity for the Aramé, Nori, and Fucus extracts and purified fractions at 0.5 and 1 mg/mL. Letters a–e correspond to values that are statistically different between the samples under study (*p* < 0.05). * results reported in [13].

**Figure 4 antioxidants-12-00684-f004:**
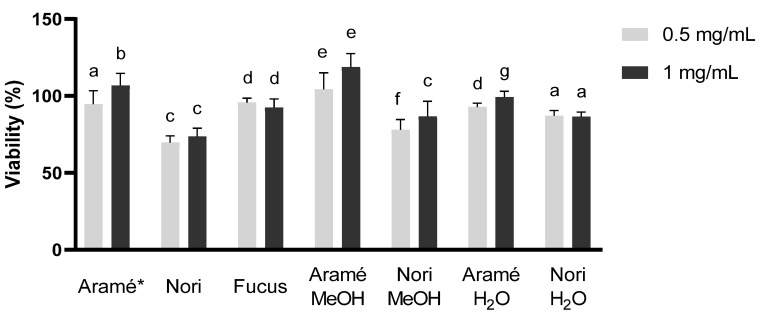
Hep-G2 cell cytotoxicity for the Aramé, Nori, and Fucus extracts and purified fractions at 0.5 and 1 mg/mL. Letters a–g correspond to values that are statistically different between the samples under study (*p* < 0.05). * results reported in [13].

**Figure 5 antioxidants-12-00684-f005:**
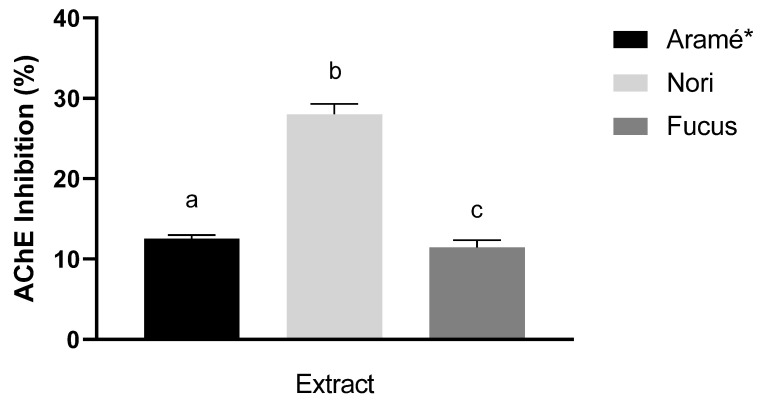
AChE Inhibitory activity for the seaweed extracts 1 mg dry mass/mL. Letters a–c correspond to values that are statistically different between the samples under study (*p* <0.05). * results reported in [13].

**Table 1 antioxidants-12-00684-t001:** Antioxidant activity of the extracts and fractions at 0.25 mg/mL. Letters a–d correspond to values that are statistically different between the samples under study (*p* < 0.05). * results reported in [13].

Aramé	Nori	Fucus	Aramé MeOH	Nori MeOH	Fucus MeOH	Aramé H_2_O	Nori H_2_O	Fucus H_2_O
65.1 ± 2.7 ^a *^	2.0 ± 0.4 ^b^	20.8 ± 0.1 ^c^	17.4 ± 1.0 ^c,d^	2.0 ± 1.6 ^b^	7.8 ± 1.0 ^d^	9.8 ± 1.4 ^d^	4.0 ± 0.9 ^b^	7.6 ± 0.04 ^d^

**Table 2 antioxidants-12-00684-t002:** Tentative identification of several compounds presents in the extracts of Aramé, Nori, and Fucus MS fragmentation pattern, Molecular Formula, and corresponding heatmap representing the intensity of the compounds in different extracts. The color gradient of the heatmap illustrates different intensities between 19812 and 908170, from lighter to darker blue.

*Rt* (min)	[M-H]^−^ *m*/*z*	MS Fragmentation	Molecular Formula	Extracts			Tentative Identification
				Aramé	Nori	Fucus	
2	191.0196	57.03 (100%); 87.00 (86.0%); 111.00 (53%)	C_6_H_8_O_7_				Citric acid [8]
	551.1821	505.18 (100%); 59.01 (59.5%);179.06 (34.8%); 71.01 (15.6%); 101.02 (10.4%)	C_19_H_36_O_18_				Glycidil compound [55]
5.9	575.0126	495.06 (81.6%); 233.05 (59.9%); 139.00 (13.9%); 261.00 (13.5%); 125.02 (10.2%); 111 (10.1%)	C_30_H_24_O_12_				Eckol derivative
7.3	596.0338	261.00(100%); 369.02 (36%); 139.00 (33.2%); 125.02 (27.4%); 111.00 (18.9%); 556.06 (13.3%)	C_30_H_28_O_13_				Eckol derivative
13.2	265.1482	96.96 (100%); 79.96 (32.8%)	C_12_H_10_O_7_				Fuhalol derivative
14	311.1691	183.01 (47.6%); 79.96 (28.6%); 119.05 (27.8%)	C_19_H_20_O_4_				Phloroglucinol derivative (Fucol type)
15.1	325.1841	183.01 (42.6%); 119.05 (32.6%);79.96 (29%)	C_19_H_18_O_5_				Phloroglucinol derivative (Fucol type)
16.3	339.2002	183.01 (34%); 119.05 (23.5%); 79.96 (18.9%)	C_19_H_16_O_6_				Phloroglucinol derivative (Fucol type)

## Data Availability

Not applicable.

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
