# Peer review of "Edible Seaweeds Extracts: Characterization and Functional Properties for Health Conditions"

_antioxidants, 2023, doi:10.3390/antiox12030684_

Round 1

Reviewer 1 Report

The manuscript ''Edible seaweeds extracts: characterization and functional properties for health conditions'' by Mariana et al., is a result of a well conducted research work but unfortunately lack of novelty.

The main aim of this research work was to compare the bioactive potentials between three seaweeds such as Aramé, Nori and Fucus. But already many research work has been done previously on these particular species and topic, thus the present work does not come up with any novel approach or significant findings.

The experimental approach or design can be improved/changed to modify the manuscript further and make it more sensible for the readers. For example, based on the results, it is stated in the manuscript that whole extracts performed well compared to the fractions when its comes to bioactivity. Thus it could be quite possible that bioactivity may depend on combinations of compounds present in whole extract which are obviously missing in the fractions. I would definitely recommend to use GNPS based networking approach with the LC-MS data set to further characterize compounds present in whole extract and fractions.

Reviewer 2 Report

The authors of the manuscript antioxidants-2191992 characterize and analyze the activities of three seaweed species, Eisenia bicyclis, Phorphyra tenera, and Fucus vesiculosus, with increased popularity as healthy food. The manuscript has strengths and weaknesses.

My ethical concern is related to the use of hepatocellular carcinoma cell line Hep-G2 (ECACC#85011430) and colorectal adenocarcinoma cell line Caco-2 (ECACC#86010202) for cytotoxicity of the extracts. The authors found that the aqueous extract of Phorphyra tenera  (1 mg/ml) stimulated Caco-2 cells, and the fractions from Eisenia bicyclis stimulated Hep-G2 cells. The authors did not consider the cancerous nature of the cells used for their tests. The extracts that stimulate cancer cells are considered safe? The conclusion is that these seaweeds, which stimulate hepatic and colorectal cancer cell proliferation, are health-promoting food.

ISO 10993 is a series of standards that evaluate the biocompatibility of medical devices. Usually, the threshold established by ISO 10993-5 for cytotoxicity of medical devices is extended to dietary supplements and functional foods – for research purposes only! The use of carcinoma cell-line in cell testing was promoted by their ability to proliferate – because they are cancer cells with modified physiology and repressed apoptosis genes. In the last years were developed immortalized primary cells (e.g., Human telomerase reverse transcriptase (hTERT)– immortalized primary cells) that combine extended proliferative capacity with the physiology of a normal cell. These are the recommended cells to test dietary supplements and functional foods.

The Caco-2 and Hep-G2 cells are used to test the plant extracts that have anti-cancer activity. I randomly present several papers from the last decades: Popovich, D. G., & Kitts, D. D. (2004). Mechanistic studies on protopanaxadiol, Rh2, and ginseng (Panax quinquefolius) extract induced cytotoxicity in intestinal Caco‐2 cells. Journal of biochemical and molecular toxicology, 18(3), 143-149; Baenas, N., Silván, J. M., Medina, S., de Pascual-Teresa, S., García-Viguera, C., & Moreno, D. A. (2015). Metabolism and antiproliferative effects of sulforaphane and broccoli sprouts in human intestinal (Caco-2) and hepatic (HepG2) cells. Phytochemistry Reviews, 14(6), 1035-1044; Rahal, N. B., Barba, F. J., Barth, D., & Chevalot, I. (2015). Supercritical CO2 extraction of oil, fatty acids and flavonolignans from milk thistle seeds: Evaluation of their antioxidant and cytotoxic activities in Caco-2 cells. Food and Chemical Toxicology, 83, 275-282; Efenberger-Szmechtyk, M., Nowak, A., & Nowak, A. (2020). Cytotoxic and DNA-damaging effects of Aronia melanocarpa, Cornus mas, and Chaenomeles superba leaf extracts on the human colon adenocarcinoma cell line Caco-2. Antioxidants, 9(11), 1030; Janatová, A., Doskočil, I., Božik, M., Fraňková, A., Tlustoš, P., & Klouček, P. (2022). The chemical composition of ethanolic extracts from six genotypes of medical cannabis (Cannabis sativa L.) and their selective cytotoxic activity. Chemico-Biological Interactions, 353, 109800.

Reviewer 3 Report

Reviewer comments and suggestions

Manuscript ID: antioxidants-2191992

With this manuscript, the authors studied and evaluated the use of 3 seaweed natural extracts, in health improvement and management of diseases, mainly those associated with oxidative stress and neurodegeneration.  The development of novel and natural products with the incorporation of these seaweeds into the diet is very important for consumers and stakeholders.

In my opinion, the manuscript is suitable for publication in Antioxidants, after revision, because the importance, innovation, and some aspects of the paper are not totally clearly demonstrated.      

Some concrete comments would be as follows:

- The abstract must clearly state the originality of this study.

- How could this study contribute positively to consumers and stakeholders?

- Did the authors determine any sensory characteristics of the obtained extracts? What was their final color?

- The authors must clearly state in the introduction the importance of this study because the application and bioactive composition of these extracts are investigated to a great extent.

- In what way is this paper differentiated from the others?

-  The authors should explain why they chose these seaweeds. What are the benefits of this kind of species? Are there used in other types of food?

-  The authors must explore the idea (lines 449-452): “As showed by Brand-Williams and colleagues [67] there is a correlation between the interaction with DPPH radical antioxidant capacity and the structural conformation of the compounds, therefore differences in the extracts composition may explain this behavior.”

- What is the influence of the seaweed extract color on the performance of the analytical methods used?

- Was any experiment design used?  

- Did the authors investigate/study the influence of geographical and environmental factors of seaweeds in the bioactive compound content? This is a very important issue.

- The authors should address future trends and perspectives.

- The validation of the conclusions must be strongly emphasized with a broader discussion around this issue.

-  Ensure that all references are the most recent and relevant to the arguments in the paper. 

In summary, I recommend this manuscript for publication in Antioxidants, but after making all corrections suggested in reviewed version of the manuscript.

Round 2

Reviewer 1 Report

I am quite satisfied with most of the justifications given by the authors against the questions raised me on previous version of the manuscript. But still I expect further clarification on probable compounds detection that was done in the manuscript. As authors mentioned the justification that 'The compounds were tentatively identified based on the MS fragmentation patterns, literature sources including AA group previous work, molecular masses and predicted molecular formulae.' -  I  would suggest them to add some probable reference structures of the compounds that they mentioned in the table 2. 

Reviewer 2 Report

The authors of the manuscript antioxidants-2191992 made some improvements to their manuscript. However, their experimental approach for testing the seaweed extracts safety still has weaknesses. They mentioned in their answer that the Caco-2 cell system is a “surrogate for human intestinal permeability measurements by the US FDA to support new drug applications”. Extracts from seaweeds are not drugs – these extracts could be considered food supplements. The application was related to safety and not to intestinal permeability.

In their modified manuscript, they cited a review, Arzumanian, Viktoriia A., Olga I. Kiseleva, and Ekaterina V. Poverennaya. "The curious case of the HepG2 cell line: 40 years of expertise." International journal of molecular sciences 22.23 (2021): 13135. In this cited review, the final table (Table 3) includes data suggesting that the “HepG2 cell line may not be a suitable model in investigating metabolism-mediated toxicity”, i.e., safety tests. To avoid reader misleading, authors must include at least one paragraph related to the limits of their safety assay in the Discussion section.

Sub-section 3.3 must be carefully verified and corrected. Since the “The compounds were tentatively identified based on the MS fragmentation patterns, literature sources [8, 53, 54], molecular masses and predicted molecular formula,” authors must correct expressions such as “phlorotannins, eckol and phloroeckol, fuhalol and other phloroglucinol derivatives were identified (L397-398)” –  must be “tentatively identified”.

I suggested moving Figure 6 to Supplementary material as a higher quality image and adding the structural formula of the proposed compounds and the MS fragmentation patterns.

The abstract must be reduced to 200 words.

Reviewer 3 Report

The authors replied to my comments and they have provided a new and improved version of the paper.

In my opinion the manuscript is suitable for publication in Antioxidants.

Round 3

Reviewer 1 Report

I did not find any attached file related to authors response towards my previous question asked. Either its technical issue or authors should make sure they have attached/submitted all the required files.

Reviewer 2 Report

The authors made the improvements. Manuscript could be published in the present form.